# Polarons shape the interlayer exciton emission of MoSe₂/WSe₂ heterobilayers

Pedro Soubelet [1] ✉, Alex Delhomme[1], Elena Blundo [1,2], Andreas V. Stier [1] & Jonathan J. Finley [1,2] ✉

We present evidence for the strong participation of hot phonons in the photophysics of interlayer excitons (IXs) in $2H-$ and $3R-$ stacked MoSe₂/WSe₂ heterobilayers. Photoluminescence (PL) excitation spectroscopy reveals that excess energy associated with relaxation of intra-layer excitons towards IXs profoundly shapes the overall IX-PL lineshape, while the energy of the spectrally narrow discrete emission lines conventionally associated with trapped moiré IXs remain unaffected. A strikingly uniform line-spacing of the discrete emission lines is observed, along with characteristic temperature and excitation level dependence. Results suggest an entirely new picture of the discrete IX emission in which non-thermal phonons play a crucial role in shaping the spectrum. Excitation power and time resolved data indicate that these features are most likely polaronic in nature. Our findings extend the understanding of the photophysics of IXs beyond current interpretations based primarily on moiré-trapped IXs.

Electron-phonon (e-ph) interactions play a major role in the charge and exciton dynamics of bulk-semiconductors, where they dominate carrier transport and optical properties[1–3]. In particular, optical phonons are able to couple with electronic states, forming polarons that transform the optical susceptibility spectrum[1–3]. For example, e-ph interactions modulate the lineshape of interband emission spectra via the emergence of phonon sidebands[4–8]. These effects are expected to be particularly strong in two-dimensional (2D) semiconductors, such as direct gap monolayer (ML) semiconducting transition metal dichalcogenides (TMDs), due to weak dielectric screening that also result in strongly bound excitons[9–17]. Experimental and theoretical investigations of the impact of polarons in 2D systems are surprisingly scarce[1,18–20], although the fingerprints of polarons have been reported in both Raman[21,22] and ARPES[23] experiments. The formation of polarons require that excitons and phonons spatially overlap, with a phonon displacement able to modulate the exciton wavefunction. Consequently, in monolayer TMDs, excitons couple to longitudinal acoustic modes at the Brillouin zone edge, with characteristic energies of ~30 meV[23].

When two different TMD monolayers are vertically stacked to form van der Waals heterobilayers (HBs) their excitonic photophysics

becomes even richer[24,25]. Interlayer hybridization gives rise to an in-plane periodic moiré potential with a periodicity determined by the twist angle ($\theta$) and the respective lattice parameters of the constituent monolayers[26–28]. This results in folding of the acoustic phonon branches leading to moiré phonons[29–32]. Moreover, for HBs having type-II band alignment, such as tungsten diselenide (WSe₂) and molybdenum diselenide (MoSe₂), the lowest energy excitons form via charge transfer between the layers. They are, therefore, spatially indirect interlayer excitons (IXs), with conduction band electrons primarily located in the MoSe₂ layer and valence band in the WSe₂[33]. The periodic moiré potential also confines IXs in the plane of the HB, facilitating the study of strongly correlated quantum phases and opening the path for novel photonic applications[34–38,38–45].

Since IXs typically lie several hundred millielectronvolts lower in energy than their intra-layer counterparts, their formation involves the absorption and emission of phonons to dissipate the excess photon energy.[46–53] Hereby, a non-equilibrium phonon population is produced by near resonant excitation of intra-layer excitons[46–49,51,52,54,55]. Moreover, IX formation modifies the equilibrium separation between the constituent TMD layers[56,57]. Theoretical predictions suggest that electron-phonon interactions then couple the relative motion of the

[1]Walter Schottky Institut and TUM School of Natural Sciences, Technische Universität München, Am Coulombwall 4, Garching, Germany. [2]Munich Center for Quantum Science and Technology (MCQST), Munich, Germany. ✉e-mail: pedro.soubelet@wsi.tum.de; jj.finley@tum.de

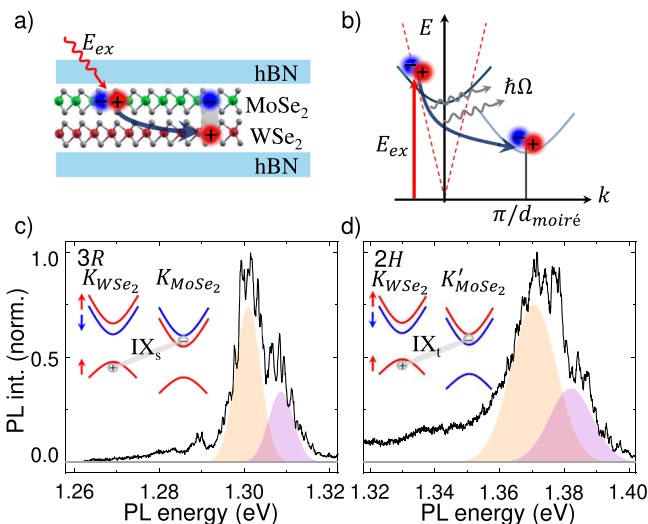

**Fig. 1 | IX formation and emission.** Schematic of the investigated hBN-encapsulated WSe₂/MoSe₂ HBs. Sketch of the IX formation in real (**a**) and momentum (**b**) space. An electron-hole pair is excited via the absorption of a photon with energy $E_{ex}$ (red arrow) within the light cone (red dotted lines). During the charge transfer and energy relaxation process, the electron-hole pair strongly interacts with the lattice by absorbing and emitting phonons ($\hbar\Omega$). **c**, **d** show the normalized IX emission intensity for the 3R−and 2H−samples, respectively. The general shape of each spectrum exhibits two lobes (orange and violet shaded regions). Insets in (**c**, **d**): Schematics of the spin-valley configuration for the 3R−and 2H−sample. The corresponding excitons are indicated.

## Results

To investigate the physics governing the IX formation and emission, we fabricated HBs using the well-known tear-and-stack method[58] to assemble twisted bilayers with near 2H ($\theta$ ~4. 5°) and 3R ($\theta$ ~-57°) twist angles. These angles were chosen to maintain significant oscillator strength while avoiding lattice reconstruction effects that arise at small $\theta$[59–63]. Figure 1a shows schematic representation of the hexagonal boron nitride (hBN) encapsulated HBs used in this work. Further details on the fabrication and characterization of the samples can be found in section IV and the Supplemental Note (SN) 1. ("*See the Supplementary Information for further details.*") Despite local variations across the samples are present, all key findings pertaining to IX-polaron physics were reproducible at multiple locations on each sample and remained unaffected by the twist angle.

The IX formation in real and momentum space is depicted in Figures 1a, b, respectively. It involves the absorption of a photon of energy $E_{ex}$ within the light cone (red dashed line in Fig. 1b) that forms into an intralayer electron-hole pair in one of the component monolayers. The charge transfer between MLs occurs over fs-timescales through the $\Sigma − (\Gamma −)$ hybridized states of the WSe₂(MoSe₂) Brillouin zones[47,50,53]. This process generates phonons ($\hbar\Omega$) via the dissipation of several hundreds millielectronvolts of energy as each *intra*-layer exciton transforms into an IX[46–49,51,52,54]. In particular, the charge transfer process results in the excitation of the interlayer breathing mode, with an energy of approximately 3.3 meV[55,64,65]. Note that the IX is a spatially and momentum-indirect state, since its $k −$ vector is situated at the edge of the mini-Brillouin zone ~ $\pi/d_{moiré}$, where $d_{moiré}$ is the moiré periodicity (see Fig. 1b)[47,66]. Therefore,

the interaction with the lattice is expected to strongly impact on the radiative recombination of IXs, as investigated in detail below.

Typical photoluminescence spectra recorded from the HBs are presented in Figure 1c, d and display a lineshape consistent with other reports in the literature[40,67–73]. The low temperature (4.2 K) PL spectra for the 3R and 2H samples in figure 1c and d were obtained using weak resonant optical pumping (excitation power $P_{ex}$ ~100 nW) of the MoSe₂ intralayer exciton at $E_{ex} = 1.602$ eV and 1.616 eV, respectively. The resulting emission stems from the singlet(triplet) IX configuration for the 3R(2H) sample[71,72]. Details regarding the exciton complexes and selection rules are presented in Supplemental Note 1. Typically, IX emission lineshapes are composed of multiple peaks, the details of which depend significantly on the experimental conditions. However, ~10 meV broad lobes have been attributed to motionally quantized states within the moiré potential (as indicated by the orange and violet shades in Fig. 1c, d)[66,74–77]. On top of these broader emission features and specifically at low to medium excitation powers, a number of sharper emission lines (sub meV linewidth) are superimposed. These lines have been frequently ascribed to the emission of individual or few IXs confined at individual moiré sites that, due to local sample inhomogeneities, lead to emission at varying energy[46,72,78]. However, we continue to present compelling evidence that the sharp emission lines as those observed in Fig. 1c, d arise from hot polarons generated during the IX generation process. Note that at very low excitation power (tens of nano Watts), even narrower emission lines ( ~100 $\mu$eV of linewidth) have been observed. This different kind of emission has been shown to exhibit single-photon emission characteristics[79] or have recently been interpreted as interlayer donor-acceptor emission centers[80].

### PL excitation and phonon participation in the IX emission

To show that phonons play a central role to the IX photophysics, we performed PL excitation (PLE) experiments, varying the excitation photon energy $E_{ex}$ across the MoSe₂ and the WSe₂ intralayer exciton. As such, we tune the excess energy. This energy ($\Delta = E_{ex} − E_{IX}$ where $E_{IX}$ is the IX emission energy) is dissipated during the IX formation process. Figure 2a shows false color PLE maps for each twisted HB sample. The narrow IX emission features remain independent of $E_{ex}$ and indicates that their energy only depends on the IX recombination process and is insensitive to the IX formation process (electron-hole excitation, charge transfer and relaxation processes). On the other hand, both samples display an enhancement of the IX emission intensity when the intralayer excitons are pumped resonantly (horizontal dotted lines). The general lineshape of the IX emission depends on $E_{ex}$ as the high energy lobe displays a resonance that is slightly blueshifted with respect to the intralayer excitons. This is highlighted by the selected spectra presented in Figure 2b that show the IX-PL recorded with $E_{ex}$ ~20 meV above (blue) and ~10 meV below (green) the intralayer exciton resonances. Each spectrum is plotted on a relative energy axis with respect to the central dip in the overall spectrum between the broad emission lobes. Exciting the HBs at an energy above the intralayer exciton resonance generally enhances the prominence of the high energy lobe, as compared to the low energy lobe. Similar PLE experiments were repeated at higher $P_{ex}$ and this behavior remained consistent (see SN 2 for details).

Remarkably, careful inspection of the sharp emission lines in Fig. 2b shows that the overwhelming majority of observable peaks are uniformly spaced in energy. Figure 2c presents the energy spacing versus an integer $N$, where $N = 0$ is chosen to correspond to the emission line in between the two emission lobes. The constant spacing between lines of ~0.8 meV cannot be explained with local moiré potential variations[72], or filling of a single moiré trap with dipolar excitons[81], but rather points towards a specific phonon mode modulating the spectral properties of the IX emission. Moreover, this even distribution of peaks in the PL spectra was observed across multiple spots on each sample studied (see SN 3).

layers, described by acoustic flexural phonon modes that couple to IXs, leading to the formation of IX-polarons[56,57]. In this letter, we provide compelling experimental evidence that spectrally narrow IX emission lines in MoSe₂/WSe₂ HBs, conventionally ascribed to the recombination of individual moiré-trapped IXs, may also arise from IXs dressed with an integer number ($N$) of non-thermal acoustic phonons.

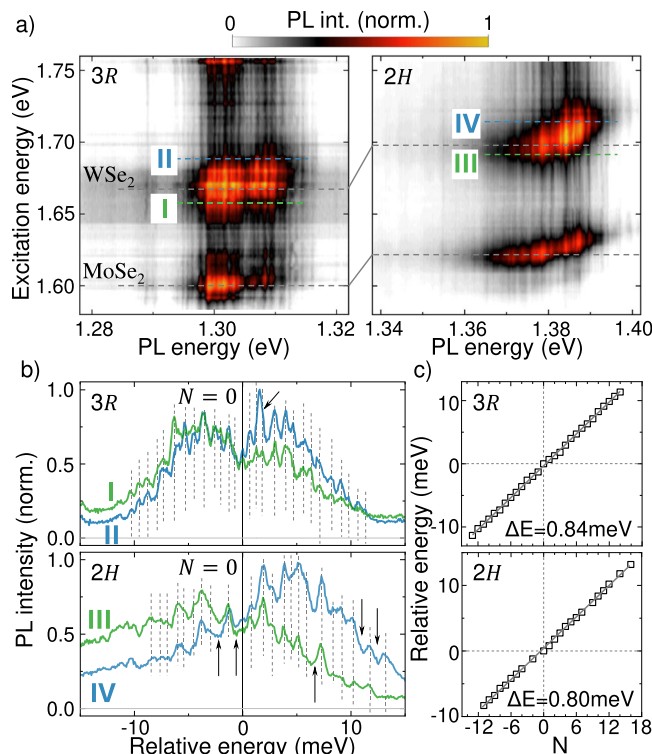

**Fig. 2 | Excess energy and polaronic nature of the IX emission. a** False color plot of the IX PL spectra for a series of excitation energies $E_{ex}$. Horizontal gray dashed lines correspond to the resonant excitation of MoSe$_2$ and WSe$_2$ intralayer excitons. **b** Selected IX PL spectra of the 3R−(top) and 2H−(bottom) samples, plotted on an energy scale relative to the local minima of the spectra in between the emission lobes. The $N = 0$ peak is at  -1.305 eV and  -1.380 eV for the 3R and the 2H sample, respectively). Spectra in green correspond to the IX emission obtained by exciting the sample just below the intralayer exciton of WSe$_2$ at $E_{ex} = 1.653$ eV (1.687 eV) for the 3R−(2H−) sample (green dotted lines in a) and the blue spectra are excited just above the resonance, at $E_{ex} = 1.685$ eV(1.720 eV) for the 3R − (2H − ) sample (blue dotted lines in (**a**). Vertical dotted lines mark the position of the sharp PL features. Black arrows mark the position of peaks that were not discernible. **c** Peak energy as function of peak number for the 3R−(top) and the 2H−(bottom) sample. The peaks are uniformly spaced with an energy interval of  -0.8 meV. Energy error bars are below the dot size.

## Temperature evolution of the IX emission

To characterize the impact of phonons on the IX emission, we performed temperature ($T$) dependent PL using $P_{ex} = 300$ nW and $E_{exc} = 1.96$ eV. Representative spectra recorded from the 2H − sample are presented in Fig. 3a. Data recorded from the 3R−sample is presented in the supplemental note 4. All spectra are normalized to the intensity of the $T = 7.2$ K spectrum. The vertical gray dotted line in the figure marks the energy of the central dip between lobes. With increasing temperature, the high energy lobe rapidly decreases in intensity, while the low energy lobe initially maintains its intensity and then decreases above a temperature of  -20 K. Figure 3b shows the temperature dependent integrated intensity of each lobe in the PL spectrum for both samples. The narrow emission lines progressively fade out, and become indistinguishable from the broad background above 30 K. Importantly, we note that they do not shift in energy throughout this range of temperature. While increasing the temperature above 35 K, the global emission redshifts and successively decreases in intensity, reaching  -5% of its low temperature peak by  -45 K.

Inelastic light scattering by phonons, such as Raman scattering or phonon cascade processes[82], are incoherent phenomena where the scattering cross-section is proportional to the thermal occupation number of phonons. Consequently, as the temperature increases, the

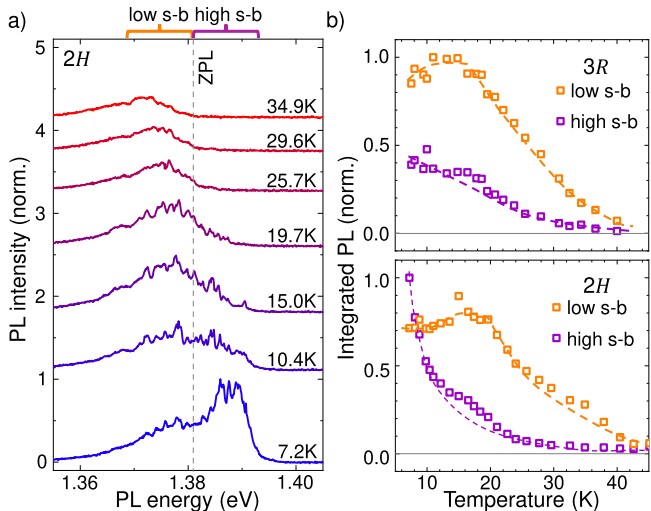

**Fig. 3 | Temperature evolution of the phonon sidebands in the IX emission. a** Temperature dependent PL spectra for the 2H−sample. On top, the region of ± 12 meV from the ZPL is marked with orange(violet) to define the low(high) energy sideband. **b** PL intensity integrated on the spectral regions defined in a. In orange(violet) is the low(high) energy sideband intensity. Upper panel correspond to the 3R − sample and the bottom panel to the 2H−sample. Error bars are below the dot size.

intensity of the scattered light in the spectra also increases. In contrast, the fade-out of the phonon replicas with increasing temperature is a fingerprint of polaronic states which dephase at elevated temperatures[4,22]. However, the presence of multiple lobes in the emission spectrum (Fig. 1c, d) is not a typical feature associated with polaron behavior. In most semiconductors, the vibrational modes that lead to prominent phonon sideband emission are typically optical modes with energies $\hbar\Omega \gg k_B T$, where $k_B$ is the Boltzmann constant[4,8]. Consequently, the population of that phonon at thermal equilibrium is strongly suppressed and the polaron formation is dominated by spontaneous emission processes, i.e. those processes in which the exciton loses energy by emitting phonons, leading to a redshifted phonon sideband. Nonetheless, the spacing we observe in the PL spectra is $\hbar\Omega$ -$k_B T$ and, therefore, absorption processes in which the interaction with the lattice increase the exciton energy and give rise to a blueshifted phonon sideband must be considered. Our observations are consistent with a MoSe$_2$/WSe$_2$ IX emission that is shaped by phonon sidebands. In this picture, the central dip in the overall spectrum between the broad lobes would correspond to the zero phonon line (ZPL) and the orange and violet lobes in Fig. 1c, d correspond to processes in which the IX energy is reduced or increased by the emission or absorption of acoustic phonons forming IX-polarons[56,57]. The sharp PL lines are phonon replicas from IX-polarons dressed with an integer number of $N$ − phonons.

The charge transfer and energy relaxation processes increase the phonon population[46–49,51,52,54,55]. Therefore, populations of IXs and phonons are out of thermal equilibrium and spatially well-overlapped, enhancing their interaction probability. In this picture, the observed $T$ − dependence of the high-energy sideband arises from processes in which phonons are annihilated. Since the phonon limits the lifetime of the IX-polaron, upon increasing temperature, the phonon lifetime reduces monotonically[83,84] and the IX emission intensity of the high energy sideband is expected to reduce accordingly. In contrast, the low energy lobe is composed of additional spontaneous emission of phonons, a process that is temperature independent. Consequently, it maintains its intensity until a threshold temperature, above which the thermal energy of the crystal affects the radiative lifetime of the IX. This behavior is precisely what is observed in Fig. 3b.

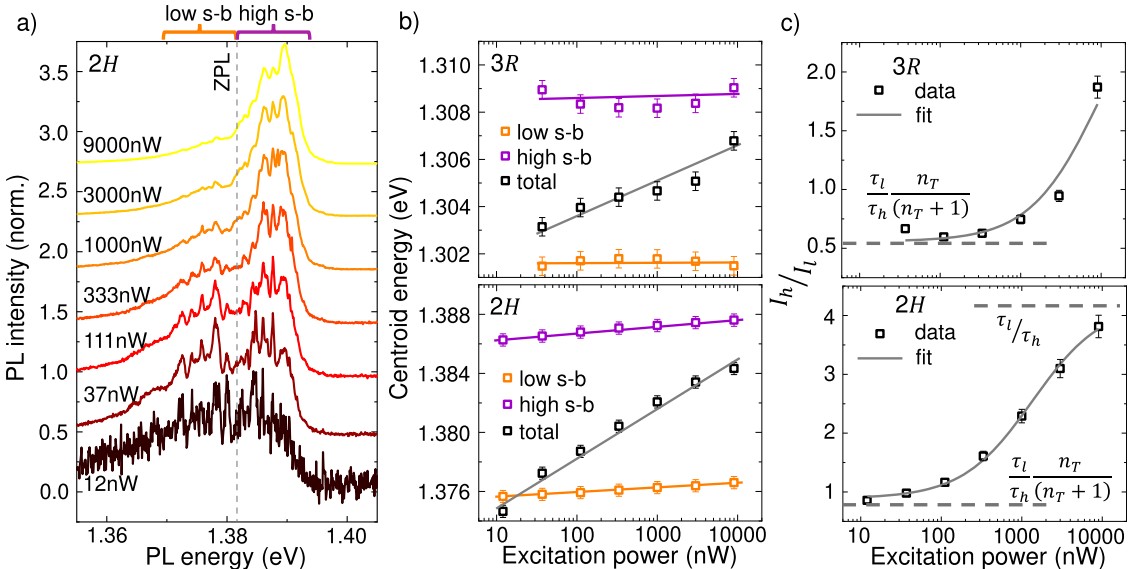

**Fig. 4 | Excitation power dependence of the phonon sidebands in the IX emission. a** Power dependent PL spectra for the 2*H*−sample. The region of ± 12 meV around the ZPL is marked with orange(violet) to define the low(high) energy sideband. **b** Centroid energy of the total spectra (black) and the low and high energy sideband in orange and violet, respectively. **c** Integrated intensity of the high energy sideband divided by the integrated intensity of the low energy sideband for the 3*R* − (top) and the 2*H* − (bottom) sample as a function of $P_{ex}$. The transition from a low to a high power regime can be modeled with a rate equation model (solid gray).

The electron-phonon coupling strength is usually derived from the intensity distribution between the different phonon replicas and it is parameterized via the Huang-Rhys factor[85–87]. By performing multi-peak fit to the spectra presented in Fig. 2b we obtained Huang-Rhys factors of 3.1 ± 0.4 and 1.6 ± 0.2 for the samples stacked in the 3*R* and 2*H* configurations, respectively. The estimation of Huang-Rhys factors greater than one for both samples is indicative of strong e-p coupling, supporting our identification of IX-polarons as determining the underlying photophysics. However, the variation in lineshape across the sample suggests that these values will be influenced by the local environment at different points on the sample. Additional details of the Huang-Rhys factor estimations can be found in SN 3.

**Non-thermal phonons and power dependence of the IX emission**
Further support for the non-thermal phonons generated during the IX formation process and the polaronic origin of the periodic sharp line emissions is obtained from power-dependent and time-resolved spectroscopy. Figure 4a shows the power-dependent PL recorded at 7 K from the 2*H*−sample for an excitation power $P_{ex}$ increasing from 12 nW to 9 μW, with a CW laser at $E_{ex}$ = 1.96 eV. Similar data from the 3*R*−sample is presented in SN 5. In both cases, the sharp polaron peaks coalesce into a broad emission line with increasing $P_{ex}$. Importantly, in the power range in which the sharp lines can be discerned, the energy of the narrow emission lines does not blueshift, similar to the absence of temperature dependent shifts discussed above. In addition, there is a clear blueshift of the overall emission upon increasing $P_{ex}$. This apparent blueshift of the overall IX emission has been widely interpreted in the literature as arising from dipolar IX-IX interactions[40,46,72,88]. Figure 4b presents the centroid energy of the recorded spectra. The upper and lower panels correspond to the 3*R*−and the 2*H*−samples, respectively. The black square symbols show, in both cases, the blueshift of the total IX emission centroid. However, this observation is completely different when we consider the centroid of each sideband, presented in orange(violet) dots for the lower(higher) energy sideband. This analysis reveals that each sideband is spectrally fixed, and the observed apparent blueshift arises mainly from an increasing dominance of the high energy sidelobe as compared to the low energy sidelobe. The lack of an actual blueshift in the

IX emission is consistent with recent literature that takes into account a more comprehensive view on IX interactions beyond simple dipolar repulsive interactions[89].

We qualitatively account for the observed power dependence using a simplified phenomenological model subject to the following assumptions: i) The phonons contributing to the sidebands are generated during the charge transfer and energy relaxation processes. Consequently, the phonon mode occupation number is $n = n_T + n_P$, where $n_T$ is the thermal occupation given by the Bose-Einstein distribution and $n_P \propto P_{ex}$ is the optically generated number of phonons. ii) The intensity of the $N^{th}$ emission line corresponds only to $N$ phonon absorption for $N > 0$ and $N$ phonon emission for $N < 0$. Under these approximations, the PL intensity of the $N = 1$ peak then corresponds to all IXs dresses by a single phonon, with an intensity proportional to $n$. Therefore, the intensity of the $N = 1$ peak is $I_{N=1} \propto n/\tau_{N=1}$, where $\tau_{N=1}$ is the radiative lifetime of this polaron. Analogously, the PL intensity of the $N = -1$ peak is given by $I_{N=-1} \propto (n + 1)/\tau_{N=-1}$, where $\tau_{N=-1}$ is its radiative lifetime and the factor $n + 1$ corresponds to the presence of both stimulated and spontaneous emissions. The ratio between these quantities becomes

$$\frac{I_{N=1}}{I_{N=-1}} = \frac{\tau_{N=-1}}{\tau_{N=1}} \frac{n_T + n_P}{n_T + n_P + 1}. \tag{1}$$

We now examine two interesting limits of Eqn. (1) that describe the ratio $I_{N=1}/I_{N=-1}$ for low and high $P_{ex}$, respectively:

$$\lim_{n \to 0} \frac{I_{N=1}}{I_{N=-1}} = \frac{\tau_{N=-1}}{\tau_{N=1}} \frac{n_T}{n_T + 1} \tag{2}$$

and

$$\lim_{n \to \infty} \frac{I_{N=1}}{I_{N=-1}} = \frac{\tau_{N=-1}}{\tau_{N=1}}. \tag{3}$$

At low excitation power, the intensity ratio of the low and high energy sidebands is proportional to the radiative recombination rates and the thermal occupation. In contrast, at high power, only the radiative recombination rates determine the relative intensities. Note that,

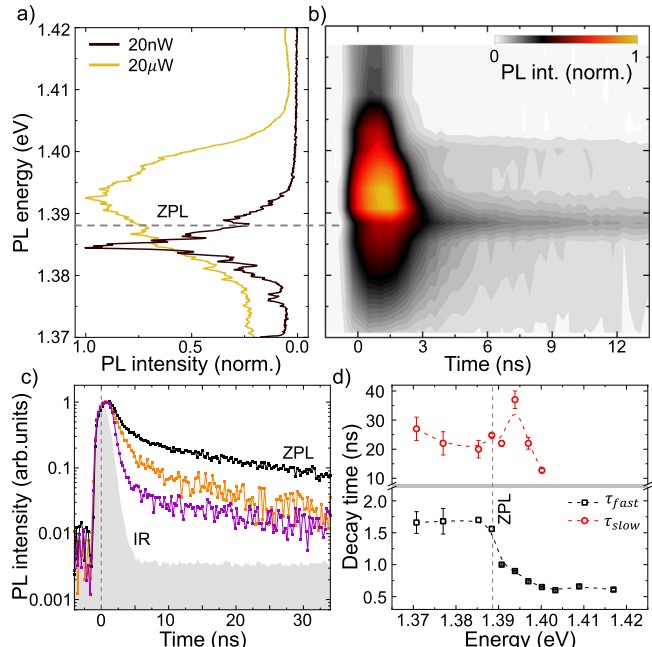

**Fig. 5 | Temporal evolution of the phonon sidebands in the IX emission. a** IX emission of the $2H-$ sample sample using a pulsed laser excitation (pulse duration 1 $ns$, $E_{ex}$ = 1.96 eV, repetition rate 1/200 ns) for average $P_{ex}$ = 20 nW and 20 $\mu W$. The ZPL is marked by a gray dotted line. **b** False color map of a time resolved IX emission spectrum for $P_{ex}$ = 20 $\mu W$. **c** PL time traces for 1.385 $eV$ (orange), 1.388 $eV$ (black, ZPL) and 1.394 $eV$ (violet). A biexponential decay is observed. In gray, the instrument response function (IR). **d** Fast (black) and slow (red) decay time obtained from a bi-exponential fit to the data in b).

strictly speaking, the intensity of the $N^{th}$ emission line is composed by absorption and emission processes, since $N = \alpha - \beta$, where $\alpha(\beta)$ corresponds to the number of absorbed(emitted) phonons[4]. Further details of the model and, particularly, the extension to $|N| > 1$ are discussed in the supplemental note 6. Without loss of generality and considering the difficulty to cleanly isolate individual polaron peaks in the spectra, we analyze the total integrated intensity of the high and low energy sideband $I_h/I_l$ as a function of power. Figure 4c shows this analysis for the $3R-$ and $2H-$samples in the top and bottom panels, respectively. In both cases, the experimental data grows from a constant value, in agreement with Eqn. (2). By increasing $P_{ex}$, the $2H-$sample shows a monotonic increase and displays a signature of saturation at a value we identify from equation (3). Despite the limitations of this model, fitting Eqn. (1) to the data of the $2H$ sample yields a ratio of the lifetimes of the lower and higher energy sidebands, $\tau_l/\tau_h = 4.2 \pm 0.1$ and a thermal occupation number $n_T = 0.27$. Remarkably, $n_T$ is consistent with a lattice temperature of 6 K, very close to the experimental conditions, providing support for the simplified model. The deduced proportionality constant between the generated occupation number $n_P$ and the excitation power $P_{ex}$ is $C = (9 \pm 1) \times 10^{-4}$ nW$^{-1}$ and, consequently, $P_{ex}$ = 10 $\mu W$ of optical power would correspond to an occupation number of $n_P \sim$10, much higher than the value in thermal equilibrium and consistent with an effective temperature of ~100 K for this particular mode (see SN 6 for details). For the $3R-$ sample, our data does not reach the limit $n \sim n + 1$ and, therefore, the fit to the data assumes $\tau_l/\tau_h = 4.2$ as in the $2H-$ sample. In this case, we obtain $n_T = 0.23$, a thermal population corresponding to 5.5 K and again in agreement with experimental conditions.

## Temporal dynamics of phonon sidebands emission

This phenomenological model predicts that the high-energy sideband would be expected to have a faster radiative decay time than the low-energy one. To test this prediction, we performed time and spectrally resolved PL spectroscopy on the $3R-$sample. The measurements were conducted at $T = 4 K$ with a diode laser at 1.96 eV, pulsewidth 1 ns and repetition rate 1/200 ns. For experimental details, see section IV. Figure 5a presents the PL spectra measured with the pulsed laser at average powers of $P_{ex}$ = 20 nW and 20 $\mu W$. As before (see Fig. 4a), there is a clear blueshift of the overall emission, but the ZPL and sharp lines observed in both spectra are independent of excitation power.

Figure 5 b presents the time-resolved PL as a false color map. The measurements were carried out at an average excitation power $P_{ex}$ = 20 $\mu W$. Clearly, the majority of the IX-emission decays in the first few nanoseconds after the excitation, while a slower decay extends to timescales of a few 10 ns. Strikingly, the ZPL is visible at ~1.388 meV, particularly in the long decay tail with higher intensity. We present time-dependent PL traces at constant energy above, below and at the ZPL in Figure 5c. A clear bi-exponential time dependence is observed, with a fast decay that depends on the PL energy. The time constants obtained from bi-exponential fits are shown in Figure 5d. These data clearly show an abrupt change of the fast decay across the ZPL, varying from ~1.6 ns below ZPL to ~0.6 ns above ZPL. Although the latter approaches our instrument response time, the ratio of the lifetimes of the lower and higher energy sidebands is $\tau_l/\tau_h \sim$2.6, in agreement with the value obtained from the power-dependent PL fits using our simplified rate equation model discussed above. This internal consistency suggests that the fast decay time of the IX emission is dominated by polaronic decay, where the non-thermal generation of phonons are due to the IX-formation process. Meanwhile, the slower time constant gradually changes across the spectral range with a peak at 1.394 eV, 6 meV above the ZPL, supporting the fact that IX-emission depends on phonons and revealing a longer intrinsic IX lifetime of ~40 ns, consistent with literature[70,90-95].

## Discussion

We now discuss which phonon modes may contribute to polaron formation. In pristine semiconductors, the vibrational modes responsible for sideband emission are typically lattice phonons[4,8,21,23]. Here, the coupling between a phonon mode and an exciton state is generally inferred from the mode atomic displacement and wavelength, which modulate the exciton wavefunction, in conjunction with the vibrational density of states at the relevant energy[21,23]. On the other hand, systems containing optically active defects, such as color centers in diamond[96,97] can also exhibit pronounced phonon sidebands. In our case, this would potentially imply that the presence of the moiré localized IX strongly distorts the lattice and decouples the phonons involved in the polaron formation from the general lattice dispersion.

Heterobilayers offer a distinct advantage for probing such vibrational modes, as the twist angle can be used to tune the phonon dispersion[29,98] and explore the mechanisms underlying IX-polaron formation. Accordingly, in Supplementary Note 7 we present the analysis of an additional sample with a twist angle of $\theta = 58.9°$. Note that the moiré lattice parameter is insensitive to the lack of inversion symmetry in the 1L-TMDs and, as a consequence, a sample with a twist angle $\theta$ is, from the moiré lattice point of view, equivalent to a sample stacked at $60° - \theta$. Therefore, the samples studied in this work have the moiré lattice parameter corresponding to twist angles $1.1°$, $3°$ and $4.5°$, covering the range of optically active samples[40,99]. Furthermore, Supplementary Note 8 details our evaluation of the phonon modes responsible for the polaronic sidebands in the IX emission, based on their dependence on the twist angle. These findings underscore the need for a deeper understanding of e-p interactions in moiré heterostructures.

In summary, we studied the photo-emission of three MoSe$_2$/WSe$_2$ HBs, with twist angles of $\theta = 4.5°$, $\theta = 57°$ and $\theta = 58.9°$. Our results strongly suggest that the narrow lines observed in the luminescence of these HBs arise from polaronic sidebands. Temperature- and power-dependent PL, and spectrally and temporally resolved PL experiments

support this conclusion. In all cases, we found compelling evidence supporting the validity of the polaronic picture. The IX formation involves a complex interplay between the optically excited electron-hole pair and the lattice, resulting in multiple phonon emission and absorption events as interlayer exciton relax to form IXs[46–49,51,52,54]. As result of these processes, IX polarizes the HB lattice and open a path to interact with low frequency vibrational modes[56,57]. Our power dependent PL suggests that by increasing the optical excitation power, the generated phonon distribution is significantly higher than the thermal population. This phonon population dominates the IX dynamics and emission at high excitation power and allows the observation of the phonon absorption sideband in the IX emission. Our results demonstrate that phonons are intricately involved with IX emission, emphasizing the need for a deeper understanding of the influence of the moiré potential on the coherent coupling between non-thermal IX and the lattice.

## Methods

### Sample fabrication

Monolayer $MoSe_2$ and $WSe_2$ were obtained from commercial bulk crystals via mechanical exfoliation. Both HBs were produced from the same monolayer $MoSe_2$ and $WSe_2$ and each monolayer was divided in two through the tear-and-stack method[58]. The HBs were stacked and encapsulated between thin hBN flakes using dry transfer techniques based on polycarbonate films, similar to ref. 100. During the stacking of each sample, the MLs were aligned to assemble samples with a known twist angle $\theta$ of approximately $4.5°$, $57°$, and $58.9°$. In $MoSe_2$/$WSe_2$ HBs stacked near $0°(60°)$, atoms within each layer try to adjust the stacking landscape forming reconstructed commensurate regions[59–63]. Therefore, we used twist angles above $2.5°$ and below $59°$ for $2H-$and $3R-$stacking, respectively, to avoid reconstruction[60–62]. For further information regarding the fabrication, characterization and stacking angle confirmation see SN 1.

### Optical experiments

PLE and time-resolved PL measurements were conducted using a helium exchange gas cryostat at $T = 4.2$ K, equipped with a cryogenically compatible objective with a numerical aperture $NA = 0.82$. For PLE measurements, we utilized a tunable Ti:Sa laser as excitation source. In time-dependent experiments, we employed a 635 nm diode laser, modulated by a TTL signal to produce excitation pulses of 1 ns duration with rise and fall times of 0.5 ns. To prevent re-excitation before the sample reaches equilibrium, the repetition rate was set to 1/200 ns. To time and spectrally resolve the IX emission, we used a monochromator fiber-coupled to a single photon detector resulting in a spectral resolution of ~0.75 meV. The time-resolved PL shown in the false color map 5b was obtained with an average $P_{ex} = 20\,\mu W$.

For temperature and power-dependent experiments, we used a helium flow cryostat with a temperature controller, enabling measurements from 7 K to room temperature. In these cases, the objective had a numerical aperture $NA = 0.60$.

## Data availability

The data supporting this study have been deposited in the NOMAD Repository under accession code https://doi.org/10.17172/NOMAD/2025.07.28-2.

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

## Acknowledgements

We gratefully acknowledge the German Science Foundation (DFG) for financial support via the SPP-2244 (DI 2013/5-1, FI 947/7-2, FI 947/7-1 and FA 971/8-1) the clusters of excellence Munich Center for Quantum Science and Technology (EXS-2111) and e-conversion (EXS-2089). P.S. acknowledges the financial support of the DFG through the Walter Benjamin program and EB acknowledges the DFG and MCQST for financial support via the distinguished postdoc program. We additionally acknowledge M. M. Glazov for the fruitful discussions on this project.

## Author contributions

P.S., A.V.S., and J.J.F. conceived the project. P.S. fabricated the samples discussed in the main text, while E.B. prepared the additional sample of SN 7. Optical measurements on the main text samples were carried out by P.S. and A.D., and thoose on the sample in SN 7 were performed by E.B. Data analysis was conducted by P.S. The manuscript was written by P.S. with input from all coauthors. All authors reviewed and approved the final version of the manuscript.

## Funding

## Competing interests

The authors declare no competing interests.
