## [Transparent Peer Review file · Nature Communications]

Polarons shape the interlayer exciton emission of MoSe₂/WSe₂ heterobilayers

Corresponding Author: Dr Pedro Soubelet

Version 0:

Reviewer comments:

Reviewer #1

(Remarks to the Author)

Soubelet et al. investigate the electron-phonon interaction in a TMD heterobilayer by reporting on the phonon sidebands of the interlayer fluorescence for different temperatures and excitation energies in the steady state as well as in the time domain. Their main result is a strong coupling to a low energy (0.8 eV) phonon, which they ascribe to a ZA acoustic mode and interpret as a signature of polaronic effects. Polaronic effects are crucial for the charge transport and optical properties as well as possible low-temperature order (charge density waves or superconductivity). Recently, evidence is building that polaronic effects are significant also in monolayer TMDs and their heterostructures with materials that show strong polaronic effects, such as organic semiconductors or titanates. The present work is the first such investigation into heterobilayers and could hence have a strong impact on our view of transport and optical phenomena in such heterostructures. The paper is well structured and of high level scholarly presentation both in the narrative and the graphics. The data is of high quality and the reasoning is well supported by arguments and references to pertinent literature. I therefore recommend publishing in Nature Communications after the following issues are addressed:

1. The interlayer breathing mode is mentioned in lines 59 and 248, stating that its energy is inconsistent with the observed line spacing. To save the reader the consultation of the given references, I suggest to put the energy of this mode in the paper
2. Since this is the first study of polaronic effects in TMD heterobilayers, the authors should elaborate more on how different or similar they are to polaronic effects in monolayers. For example, are the ZA acoustic modes expected also in monolayers and could they be excited with suitable excess energy, e.g. resonantly into the B or C exciton and observing sidebands in the A exciton fluorescence? Is such behaviour reported in the literature? If not, discuss differences to the heterobilayer case.

Reviewer #2

(Remarks to the Author)

Soubelet et al. discuss in their manuscript how polarons that are formed in the interlayer exciton (IX) formation process shape the emission lineshape of the IX PL feature. Step by step, the authors present strong evidence that the narrow linewidth features in the IX emission peak(s) are polaronic sidebands, and how the non-equilibrium excitation of the phonon system caused by the excess energy in the intralayer-to-interlayer exciton formation process shapes the overall IX peak. The manuscript is very well written and was easy to follow along, despite my lack of experience with PL measurements and spectra. I only have a few minor comments that could help to further improve the manuscript, but otherwise no major concerns about the interpretation. In my opinion, this manuscript is sufficiently important and impactful to be published in Nature Communications, as it emphasises the strong role of phonons and in particular non-equilibrium phonon occupations in the IX formation process. However, I am not so experienced what is new and what is known in the interpretation of such high-resolution PL data from IX and the observed spectrally narrow lines, so that another opinion from another reviewer might be important here.

My comments in no particular order:

- The authors tend to just write the parameters in the text (E_{ex} , P_{ex} , n_P , τ_l/τ_h , etc.) expecting the reader to remember them all. It is all very well introduced and no information is missing, but especially for the first reading it would be easier to follow along if this were improved. For example, they could write “the ratio of the lifetimes of the lower and higher energy sidebands is τ_l/τ_h ...” or “The proportionality constant between the optically generated occupation n_p and the

excitation power P_{ex} ..." etc.

- In Fig. 2a the caption does not explain the horizontal gray dashed line. Also, the E_{ex} value for the gray dashed line could be added to the y-scale, so that it is clear from the figure where the "0" in 2b can be found.
- If the higher energy sidebands would be dominantly induced by thermal occupation, wouldn't one expect that the higher-energy intensity goes up with increasing temperature at first (Fig. 3a)? I think this is another observation that directly shows that these peaks must result from a non-equilibrium occupation?
- Line 95: blueshifted with respect
- 160: capital a for additional
- 224: pulsewidth instead of pulse with
- 229: measurements "were"
- Fig. 5d: red caption in box with red data points (slow decay). Exchange fast and slow decay panel, because fast decay is discussed first?
- 223 and 224: We present... -> not only above and below, but also ZPL
- 268: ... resulting in multiple phonon...

Reviewer #3

(Remarks to the Author)

This manuscript presents an investigation of phonon replicas in interlayer exciton (IX) of MoSe₂/WSe₂ heterobilayer, emphasizing the role of polarons in multiple phonon replicas in PL spectrums. The study makes an important contribution to understanding IX dynamics and the influence of phonons. The main observation of a consistent 0.8 meV energy spacing in the emission spectrum, which is attributed to specific low-energy phonon modes, is an intriguing result that may allow a deep understanding of inter-layer exciton physics in 2D heterostructures.

However, there are significant concerns about the authors' claims and completeness of the analysis. Key aspects, such as the effect of twist angles on phonon modes, theoretical understanding of the phonon replicas, and the need for absorption spectroscopy to validate the ZPL and phonon replicas, are not adequately addressed. These issues must be resolved for publication in Nature Communications. I recommend a Major Revision.

1. The phonon modes and their interactions with excitons strongly depend on the twisted angles. However, the manuscript does not show the different twisted angle-dependent phonon replicas. More detailed experimental results and theoretical analysis are needed to support the authors' claims.
2. The authors claimed that the observation of ZPL and phonon replicas arise from specific low energy phonon interactions in twisted MoSe₂/Wse₂. However, the authors' claims are based solely on PL experiments. Absorption spectroscopy experiments are required to confirm the ZPL and phonon replica couplings directly.
3. Identifying 0.8 meV phonon mode is a very important issue in the manuscript. But still, the origin of phonon modes is not clear. The authors must show the different angle-dependent phonon mode variation experiment results and theoretical support.

Version 1:

Reviewer comments:

Reviewer #1

(Remarks to the Author)

The authors have suitably addressed my comments. I now recommend publishing of the present version of the manuscript

Reviewer #2

(Remarks to the Author)

I have carefully read and followed the changes made by Soubelet. et al. to the resubmitted manuscript and can now recommend the paper for publication.

Reviewer #3

(Remarks to the Author)

The authors' main claim is that the observed narrow emission lines, which are phonon sidebands from a low energy vibrational mode such as a folded ZA acoustic phonon. However, this phonon mode is not clearly identified, and the manuscript lacks sufficient evidence. The authors must show experimental and theoretical results to support this critical assignment.

Low energy acoustic phonon modes can vary significantly with twisted angle due to Moiré modulation and folding. So, it is essential to demonstrate the robustness of narrow phonon sideband features across different twist configurations. The authors only present two angles, and while the authors argue that this is sufficient to support their claim and establish the generality of interlayer exciton polaron shaping. It is insufficient to confirm the origin of the narrow IX phonon replicas

involved.

Without further supporting results, the interpretation of specific phonon interaction remains speculative. I suggest that the authors include at least one or two more intermediate angle differences and a more detailed theoretical analysis for low energy phonon mode and their DOS and electron-phonon coupling in twisted TMDC with twisted angles.

The lack of clarity in phonon mode identification undermines the strength of the authors' main claims. I believe that addressing this issue is essential before the manuscript can be considered for publication.

Version 2:

Reviewer comments:

Reviewer #3

(Remarks to the Author)

The authors addressed my concerns. I recommend the publication.

Point-by-point response to reviewers, Comments reproduced verbatim

Resubmission of the manuscript entitled "Polarons shape the interlayer exciton emission of MoSe₂/WSe₂ heterobilayers" by P. Soubelet et al for further consideration.

Below, we provide the referee reports (in black), together with our responses (in blue) and modifications to the manuscript (in red).

Reviewer #1:

Soubelet et al. investigate the electron-phonon interaction in a TMD heterobilayer by reporting on the phonon sidebands of the interlayer fluorescence for different temperatures and excitation energies in the steady state as well as in the time domain. Their main result is a strong coupling to a low energy (0.8 meV) phonon, which they ascribe to a ZA acoustic mode and interpret as a signature of polaronic effects. Polaronic effects are crucial for the charge transport and optical properties as well as possible low-temperature order (charge density waves or superconductivity). Recently, evidence is building that polaronic effects are significant also in monolayer TMDs and their heterostructures with materials that show strong polaronic effects, such as organic semiconductors or titanates. The present work is the first such investigation into heterobilayers and could hence have a strong impact on our view of transport and optical phenomena in such heterostructures. The paper is well structured and of high level scholarly presentation both in the narrative and the graphics. The data is of high quality and the reasoning is well supported by arguments and references to pertinent literature.

We sincerely thank the referee for these encouragingly positive remarks and detailed reading of the manuscript as well as the support for our position that the paper will appeal to a broad community.

I therefore recommend publishing in Nature Communications after the following issues are addressed:

1. The interlayer breathing mode is mentioned in lines 59 and 248, stating that its energy is inconsistent with the observed line spacing. To save the reader the consultation of the given references, I suggest to put the energy of this mode in the paper.

Reply: The breathing mode energy in heterobilayers was determined to be approximately 3.3 meV [Nano Letters 23, 8186 (2023)]. We agree with the referee that this information will be useful to readers and have incorporated this information into the text at the locations indicated by the referee.

2. Since this is the first study of polaronic effects in TMD heterobilayers, the authors should elaborate more on how different or similar they are to polaronic effects in monolayers. For example, are the ZA acoustic modes expected also in monolayers and could they be excited with suitable excess energy, e.g. resonantly into the B or C exciton and observing sidebands in the A exciton fluorescence? Is such behaviour reported in the literature? If not, discuss differences to the heterobilayer case.

Reply: We thank the referee for this enriching suggestion. For a phonon mode to efficiently couple with excitons, both particles have to overlap in the spatial and temporal domain. Moreover, the phonon wavelength should be comparable to the excitonic Bohr radius (~1-2nm). In monolayer TMDs, such modes turn out to be LA-phonons at the edge of the Brillouin zone with an energy of ~30meV. The mode

described in our work is the ZA branch at a finite q , because it efficiently modulates the spatial extent of the interlayer exciton.

To clarify this point, we have modified the introduction as follows (line 17): “Experimental and theoretical investigations of polarons in 2D systems are surprisingly scarce [1, 18-20], although the fingerprints of polarons have been reported in both Raman [21, 22] and ARPES [23] experiments. **The formation of polarons require that excitons and phonons spatially overlap, with a phonon displacement able to modulate the exciton wavefunction. Consequently, in monolayer TMDs, excitons couple to longitudinal acoustic modes at the Brillouin zone edge, with characteristic energies of $\sim 30\text{meV}$ [23].**”

- [1] Nature Physics 19, 629 (2023).
- [18] Nature materials 15, 835 (2016).
- [19] Nature communications 6, 8585 (2015).
- [20] Nature communications 7, 10386 (2016).
- [21] Nature communications 11, 4780 (2020).
- [22] Advanced Science 11, 2305182 (2024).
- [23] Nature materials 17, 676 (2018).

Reviewer #2:

Soubelet et al. discuss in their manuscript how polarons that are formed in the interlayer exciton (IX) formation process shape the emission lineshape of the IX PL feature. Step by step, the authors present strong evidence that the narrow linewidth features in the IX emission peak(s) are polaronic sidebands, and how the non-equilibrium excitation of the phonon system caused by the excess energy in the intralayer-to-interlayer exciton formation process shapes the overall IX peak. The manuscript is very well written and was easy to follow along, despite my lack of experience with PL measurements and spectra. I only have a few minor comments that could help to further improve the manuscript, but otherwise no major concerns about the interpretation. In my opinion, this manuscript is sufficiently important and impactful to be published in Nature Communications, as it emphasises the strong role of phonons and in particular non-equilibrium phonon occupations in the IX formation process. However, I am not so experienced what is new and what is known in the interpretation of such high-resolution PL data from IX and the observed spectrally narrow lines, so that another opinion from another reviewer might be important here.

We thank the referee for their thorough review of the manuscript and their overall supportive feedback.

My comments in no particular order:

1. The authors tend to just write the parameters in the text (E_{ex} , P_{ex} , n_{P} , $\tau_{\text{l}}/\tau_{\text{h}}$, etc.) expecting the reader to remember them all. It is all very well introduced and no information is missing, but especially for the first reading it would be easier to follow along if this were improved. For example, they could write “the ratio of the lifetimes of the lower and higher energy sidebands is $\tau_{\text{l}}/\tau_{\text{h}}$...” or “The proportionality constant between the optically generated occupation n_{p} and the excitation power P_{ex} ...” etc.

Reply: We thank the referee for this observation and agree that less reliance on abbreviations will improve the overall readability of the manuscript. We have incorporated his/her suggestion at several points in the text to help the reader to follow the discussion. In particular, lines: 90, 173, 220, 221, 224, 225, 241, 249, and 286.

2. In Fig. 2a the caption does not explain the horizontal gray dashed line. Also, the E_{ex} value for the gray dashed line could be added to the y-scale, so that it is clear from the figure where the “0” in 2b can be found.

Reply: We have improved the caption of Figure 2a as follows: “Excess energy and polaronic nature of the IX emission. a) False colour plot of the IX PL spectra for a series of excitation energies E_{ex} . **Horizontal grey dashed lines correspond to the resonant excitation of MoSe₂ and WSe₂ intralayer excitons.** b) Selected IX PL spectra of the 3R- (top) and 2H- (bottom) samples, plotted on an energy scale relative to the local minima of the spectra between the emission lobes. The N=0 peak is at $\sim 1.305\text{eV}$ and $\sim 1.380\text{eV}$ for the 3R and the 2H sample, respectively.)...”

For clarity we do not mark the “0” in Figure 2b, but have added the energy in the caption.

3. If the higher energy sidebands would be dominantly induced by thermal occupation, wouldn't one expect that the higher-energy intensity goes up with increasing temperature at first (Fig. 3a)? I think this is another observation that directly shows that these peaks must result from a non-equilibrium occupation?

Reply: We thank the referee for this important and insightful observation. We agree that the temperature dependence serves as a crucial experimental approach for distinguishing between incoherent and coherent processes. For example, in Raman scattering or phonon cascade processes, there is no coherence involved. As temperature increases, the phonon population also increases, leading to an enhanced scattering peak in the spectra. If the physics we observe was an incoherent process, we would observe an increase of this spectral feature with increasing temperature, simply reflecting population of the phonon mode [84]. In contrast, a polaron is a coherent quasiparticle consisting of electric charges dressed with crystal deformations. As temperature rises, the lifetime of the polaron is reduced due to the increasing contribution from decoherence processes, limiting the observation of this feature in PL. In our work, we observe the fade out of the sharp features with increasing temperature (Fig.3).

We have improved our discussion about this point in the main text as follows (line 129): “**Inelastic light scattering by phonons, such as Raman scattering or phonon cascade processes [84], are incoherent phenomena where the scattering cross-section is proportional to the thermal occupation number of phonons. Consequently, as temperature increases, the intensity of the scattered light in the spectra also increases. In contrast, the fade-out of the phonon replicas with increasing temperature is a fingerprint of polaronic states which dephase at elevated temperatures [4, 22].**”

[84] Nature communications 12, 538 (2021).

[4] Physica status solidi (b) 246, 332 (2009).

[22] Advanced Science 11, 2305182 (2024).

4. Line 95: blueshifted with respect
160: capital a for additional
224: pulsewidth instead of pulse with
229: measurements “were”
223 and 224: We present... -> not only above and below, but also ZPL
268: ... resulting in multiple phonon...
Fig. 5d: red caption in box with red data points (slow decay). Exchange fast and slow decay panel, because fast decay is discussed first?

Reply: We thank the referee for noting this. Regarding the final comment on the ordering of the panels, we would like to clarify that the figure contains a single panel with a cut axis to display lifetime dependencies across different ranges. As a result, it is not possible to invert the ordering.

Reviewer #3:

This manuscript presents an investigation of phonon replicas in interlayer exciton (IX) of MoSe₂/WSe₂ heterobilayer, emphasizing the role of polarons in multiple phonon replicas in PL spectrums. The study makes an important contribution to understanding IX dynamics and the influence of phonons. The main observation of a consistent 0.8 meV energy spacing in the emission spectrum, which is attributed to specific low-energy phonon modes, is an intriguing result that may allow a deep understanding of inter-layer exciton physics in 2D heterostructures.

However, there are significant concerns about the authors' claims and completeness of the analysis. Key aspects, such as the effect of twist angles on phonon modes, theoretical understanding of the phonon replicas, and the need for absorption spectroscopy to validate the ZPL and phonon replicas, are not adequately addressed. These issues must be resolved for publication in Nature Communications. I recommend a Major Revision.

We thank the referee for this thorough review and welcome the opportunity to revise and clarify our manuscript. Critically, we present, for the first time, an internally consistent and complete picture of the photophysics for interlayer **polarons** in heterobilayer TMDs. Our work does not only unambiguously show that phonon processes are involved, but that those phonons coherently dress the interlayer excitons. While the ZA phonon energy will depend on the twist angle between layers through the moiré wavelength, the completeness of our work, the general experimental observations and our main and novel finding will not be affected. Furthermore, we explicitly show spectral evidence of the zero-phonon line in our time resolved data. As such, we respectfully disagree with the referee on the necessity for more twist angle dependent experiments, while noting that the twist angle dependence may be subject to a follow up work in a more specialized journal in the future. Indeed, the work described by the referee is underway but will take another twelve months to realize and characterize samples, as well as performing the necessary luminescence and excitation spectroscopy. We believe that the results we present in our manuscript already strongly support our observation and would invite other authors to begin follow up studies to confirm our findings in their laboratories.

1. The phonon modes and their interactions with excitons strongly depend on the twisted angles. However, the manuscript does not show the different twisted angle-dependent phonon replicas. More detailed experimental results and theoretical analysis are needed to support the authors' claims.

Reply: As discussed above, our main claim of polaron physics dominating the interlayer exciton PL is unambiguously shown in our manuscript. Our results are strong evidence that the prevailing interpretation of interlayer exciton emission being attributed to trapped IX at different moiré sites is far from the complete picture. Thus, our finding may therefore aid in the hunt for exotic IX physics, such as condensation and it highlights the role of multi-modal couplings in the photophysics of 2D heterostructures. Furthermore, we show that the polaron physics occurs independent of the bandstructure, as we show consistent experimental observations in 2H and 3R stacked samples.

We acknowledge the referee's point that we do not unambiguously identify the origin of the specific phonon mode responsible for the polaron formation. While it lies beyond the scope of our current work,

we believe that a twist angle dependent experiment alone will not solve this issue, since low energy phonon renormalization close to $q=0$ may reshape the phonon dispersion relation [i, ii] in a spectral region that is challenging to access experimentally. Of course, 0.8 meV is below the typical range in Raman spectroscopy and approaches the higher limit of Brillouin scattering capabilities. Indeed, it is one of our hopes that our findings will stimulate such studies that would be unambiguously capable of identifying the phonons involved.

[i] ACS Nano 17, 13938–13947 (2023).

[ii] Nature Materials volume 20, 1100–1105 (2021).

2. The authors claimed that the observation of ZPL and phonon replicas arise from specific low energy phonon interactions in twisted MoSe₂/WSe₂. However, the authors' claims are based solely on PL experiments. Absorption spectroscopy experiments are required to confirm the ZPL and phonon replica couplings directly

Reply: Please, note that our time resolved data explicitly shows the ZPL spectrum.

We agree that absorption spectroscopy is a natural experiment to observe and identify electronic and excitonic states. For example, it has been successfully employed in the study of moiré *intra*-layer excitons [iii]. However, the case of *inter*-layer excitons is fundamentally different because interlayer excitons are both spatially and momentum indirect, making them inaccessible to linear absorption spectroscopy. This limitation has remained one of the key challenges in identifying and interpreting interlayer exciton emission over the years. Recently, it was shown that reconstruction effects in heterobilayers can relax the selection rules for interlayer excitons, allowing their observation via absorption spectroscopy [iv]. However, the physics governing reconstructed samples differs significantly from the focus of our current work.

[iii] Physical Review Letters 132, 076902 (2024).

[iv] Physical Review Letters 132, 016202 (2024).

3. Identifying 0.8 meV phonon mode is a very important issue in the manuscript. But still, the origin of phonon modes is not clear. The authors must show the different angle-dependent phonon mode variation experiment results and theoretical support.

Reply: As discussed in our previous reply, the key point of this manuscript is the observation and description of polaron physics governing interlayer exciton PL. Regarding theoretical support, we would like to emphasize that calculations predicting the existence of these low energy polarons are already presented in the introduction of our manuscript [57, 58].

[57] Physical Review B 105, 205305 (2022).

[58] Annalen der Physik 532, 2000339 (2020).

Finally, we sincerely thank all the referees for their careful and critical reading, which helped us identify and correct errors prior to publication. We apologize for the mistakes in the previous version of our manuscript and firmly believe that their valuable suggestions and corrections have significantly improved our work.

Point-by-point response to reviewers, Comments reproduced verbatim

Below, we provide the referee reports (in black), together with our responses (in blue) and modifications to the manuscript (in red).

Reviewer #1:

The authors have suitably addressed my comments. I now recommend publishing of the present version of the manuscript.

Reviewer #2:

I have carefully read and followed the changes made by Soubelet. et al. to the resubmitted manuscript and can now recommend the paper for publication.

We sincerely thank the referees for their positive assessments and their support for the publication of our work. We appreciate their recognition of the significance of our findings and their potential interest to a broad scientific community.

Reviewer #3:

The authors' main claim is that the observed narrow emission lines, which are phonon sidebands from a low energy vibrational mode such as a folded ZA acoustic phonon. However, this phonon mode is not clearly identified, and the manuscript lacks sufficient evidence. The authors must show experimental and theoretical results to support this critical assignment.

Low energy acoustic phonon modes can vary significantly with twisted angle due to Moiré modulation and folding. So, it is essential to demonstrate the robustness of narrow phonon sideband features across different twist configurations. The authors only present two angles, and while the authors argue that this is sufficient to support their claim and establish the generality of interlayer exciton polaron shaping. It is insufficient to confirm the origin of the narrow IX phonon replicas involved.

Without further supporting results, the interpretation of specific phonon interaction remains speculative. I suggest that the authors include at least one or two more intermediate angle differences and a more detailed theoretical analysis for low energy phonon mode and their DOS and electron-phonon coupling in twisted TMDC with twisted angles.

The lack of clarity in phonon mode identification undermines the strength of the authors' main claims. I believe that addressing this issue is essential before the manuscript can be considered for publication.

We thank the referee for the thoughtful and constructive feedback. In response to the concerns raised regarding the identification of the low-energy phonon mode and the robustness of the observed narrow phonon sidebands across different twist configurations, we have now included additional experimental data from a new sample with close to 60° twist angle (structure widely used in the literature), as well as a more detailed analysis about the nature of the low-energy phonon modes. These additions are presented as two new Supplementary Notes. We believe these new results strengthen our assignment and help clarify the physical origin of the observed features. The main text, has been modified accordingly from line 257:

Finally, we examine which phonon modes may contribute to the formation of polarons. In pristine semiconductors, the vibrational modes responsible for sideband emission are typically lattice phonons [4,8,21,23]. Here, the coupling between a phonon mode and an exciton state is generally inferred from the mode atomic displacement and wavelength, which modulate the exciton wavefunction, in conjunction with the vibrational density of states at the relevant energy [21,23]. On the other hand, systems containing optically active defects, such as color centers in diamond [98,99] can also exhibit pronounced phonon sidebands. In our case, this would potentially imply that the presence of the moiré localized IX strongly distorts the lattice and decouples the phonons involved in the polaron formation from the general lattice dispersion.

Heterobilayers offer a distinct advantage for probing such vibrational modes, as the twist angle can be used to tune the phonon dispersion [29,100] and explore the mechanisms underlying IX-polaron formation. Accordingly, in Supplementary Note VI [65] we present the analysis of an additional sample with a twist angle of $\theta = 58.9^\circ$. Note that the moiré lattice parameter is insensitive to the lack of inversion symmetry in the 1L-TMDs and, as a consequence, a sample with a twist angle θ is, from the moiré lattice point of view, equivalent to a sample stacked at $60^\circ - \theta$. Therefore, the samples studied in this work have the moiré lattice parameter corresponding to twist angles of 1.1° , 3° and 4.5° , covering the range of optically active samples [40,101]. Furthermore, Supplementary Note VII [65] details our evaluation of the phonon modes responsible for the polaronic sidebands in the IX emission, based on their dependence on the twist angle. These findings underscore the need for a deeper understanding of e-p interactions in moiré heterostructures.

We have also modified the line 279 as follows:

In summary, we studied the photo-emission of three MoSe₂/WSe₂ HBs, with twist angles of $\theta = 4.5^\circ$, $\theta = 57^\circ$ and $\theta = 58.9^\circ$.

Finally, we have removed the phrase in line 288:

~~Considering the low energy spacing between peaks, we tentatively suggest that the involved phonon belongs to the ZA phonon branch.~~

for considering it misleading considering the new Supplementary Notes.

The citations are:

- [4] T. Feldtmann, et al. Physica status solidi (b) 246, 332 (2009).
- [8] T. Feldtmann, et al. Journal of luminescence 130, 107 (2010).
- [21] W. Jin, et al. Nature communications 11, 4780 (2020).
- [23] M. Kang, et al. Nature materials 17, 676 (2018).
- [29] Z. Li, et al. Journal of Semiconductors 44, 011902 (2023).
- [40] K. L. Seyler, et al. Nature 567, 66 (2019).
- [65] Supplementary Information.
- [98] R. Brout, et al. Physical Review Letters 9, 54 (1962).
- [99] A. Zaitsev. Physical Review B 61, 12909 (2000).
- [100] L. Li, et al. Nature Communications 16, 4117 (2025).
- [101] P. K. Nayak, et al. ACS nano 11, 4041 (2017).

Below, we transcribe the additional Supplementary Notes:

VI. Supplementary note: Additional sample $\theta \approx 1^\circ$

Figure 9: **a)** Optical micrographs of the additional HB with near 2H-stacking. Green(red) lines indicate the 1L-MoSe₂(1L-WSe₂). **b)** Polar plot of the polarization-resolved SHG intensity in the monolayer region measured for each sample. The relative angle between lobes suggests a twist angle of $(58.9 \pm 0.7)^\circ$. **c)** Interlayer exciton emission at two different P_{ex} . The emission peaks are marked with dotted vertical lines and the zero order with a red line. The arrows mark emission lines that lack or spectral regions in which the lines are not evenly spaced. **d)** Peak energy as a function of peak order for the spectra in **c**.

To explore how the polaron emission bands depend on the twist angle in MoSe₂/WSe₂ HBs, we fabricated an additional sample. Increasing the twist angle above 6° (below 55°) leads to greater momentum mismatch between the electron and hole comprising the IX, leading to a HB emission up to three order of magnitude dimmer than the emission of samples with twist angles near 0° (60°) [17,27]. As it makes it challenging to resolve the narrow emission lines, we stacked a third sample with a twist angle near 60° (or, equivalently, 0°). Figure 9a presents an optical image of this sample, which was transferred onto a 270nm-thick SiO₂/Si substrate. Polarization-resolved SHG measurements [8] and PL spectroscopy confirmed a twist angle of $(58.9 \pm 0.7)^\circ$, as shown in Fig.9b and c, respectively.

Photoluminescence spectra of the sample, presented in Fig.9c, were acquired with $5\mu\text{W}$ and $10\mu\text{W}$ CW laser excitation at 720nm, resonant with the WSe₂ A-exciton. The spectra are qualitatively identical to those discussed in the main text, featuring double lobed broad emission and a sequence of narrow emission lines. A comparative analysis of the two spectra allows us to extract the peak distribution, shown in Fig.9d. Arrows in Figs.9c and d indicate emission lines that are absent or spectral regions where lines deviate from the uniform spacing. Despite minor irregularities, the emission lines exhibit an approximately uniform energy spacing of $\sim 0.5\text{meV}$. This value is significantly smaller compared to the peak spacing observed in the

samples with the larger twist angles, suggesting a subtle dependence of the polaron features on θ and highlighting the need for further understanding of the phonon mode that leads to the polaron formation.

VII. Supplementary note: Phonon modes implicated in the IX-polarons formation

In solid-state hosts, various vibrational modes can contribute to the emergence of phonon sidebands. In pristine semiconductors, these vibrational modes are usually lattice phonons [18,19,24,28]. The coupling between a phonon mode and an exciton state is generally inferred from the mode atomic displacement and wavelength, which modulate the exciton wavefunction, in conjunction with the vibrational density of states at the relevant energy [24,28].

On the other hand, systems containing optically active defects, such as color centers in diamond [29,30], can also exhibit pronounced phonon sidebands. Here, the defects induce significant distortions in the lattice crystal, giving rise to local and quasi-local vibrational modes, which are detached from the bulk phonon dispersion. In such cases, these modes exist solely due to the presence of the impurity, and their frequencies are determined not by the crystal phonon dispersion, but rather by the mass of the impurity and the local interatomic bonding forces [30]. In our case, this hypothesis suggests that the Coulomb interaction between the electron and hole that comprise the moiré-localized IX significantly perturbs the lattice, generating phonon modes that are decoupled from the lattice phonon dispersion. These local distortions could, therefore, facilitate the formation of energetically discrete and spatially localized vibrational modes that are intrinsic to the polaronic state.

In 2D materials, the unambiguous identification of the specific phonon mode involved in polaron formation is often insufficiently defined [24,28,31]. While theoretical ab-initio calculations provide the material phonon dispersion, in practice, a variety of extrinsic and intrinsic factors can affect the real phonon dispersion. Notably, phonon renormalization in 2D materials is strongly dependent on the number of layers [32,33], doping level [34], applied strain [35], nanostructured environment [36] and temperature [37]. Additionally, TMD HBs are subject to reconstruction phenomena [38], heterostrain [39] and the hybridization of vibrational states between the 1L-TMDs and with the rest of the heterostructure [40], factors that are inherently difficult to control and quantify. This complexity increases further in the case of IX-polarons [41] since the phonon modes involved in the IX-polaron formation are difficult to observe them due to their low energy.

Figure 10: **a)** Moiré lattice parameter as a function of twist angle for a MoSe₂/WSe₂ HB. **b)** Edge of the moiré mini-Brillouin zone as a function of twist angle for a MoSe₂/WSe₂ HB. Dots in **a** and **b** correspond to the investigated samples. The rectangular shades highlight the uncertainty derived from the twist angle error bar. **c)** Acoustic phonon dispersion for MoSe₂ and WSe₂ extracted from Ref. [45]. The dots superimposed on the solid lines correspond to the investigated samples. The position along the x axis of each dot is the calculated edge of the respective mini-Brillouin zone.

In this section we discuss the nature of the phonon modes that take part in the IX-polarons. The observed phonon energies of $\sim 0.5\text{meV}$ to $\sim 0.8\text{meV}$ are smaller than the interlayer breathing mode energy in $\text{MoSe}_2/\text{WSe}_2$ HBs, which is $\sim 3.3\text{meV}$ [42,43]. Therefore, if the vibrational modes belong to the 2D dispersion, they would be situated within the acoustic phonon branches, in particularly the ZA branch as proposed in Ref. [41,44]. Furthermore, the phonon involved in the polaron formation must be located at the edge of the mini-Brillouin zone ($\pi/d_{\text{moiré}}$) to absorb the momentum mismatch of the IX relative to the light cone (see Fig.1b in the main text). Here, the folded phonon dispersion flattens towards $\pi/d_{\text{moiré}}$, exhibiting a maximum in the phonon density of states that enhance the e-ph interaction probability.

Figure 10a shows the calculated moiré lattice parameter $d_{\text{moiré}}$ as a function of the twist angle for $\text{MoSe}_2/\text{WSe}_2$ HBs, calculated following Ref. [39] with a lattice parameter of $a_{\text{MoSe}_2}=0.329\text{nm}$ for the 1L- MoSe_2 [46] and 0.4% smaller for the 1L- WSe_2 [46,47]. The three dots correspond to the samples presented in this work. Note that the moiré lattice parameter is insensitive to the lack of inversion symmetry in the 1L-TMDs and, therefore, the sample stacked at $57^\circ(58.9^\circ)$ is, from the moiré lattice point of view, equivalent to a sample stacked at $3^\circ(1.1^\circ)$. The rectangular shades in Fig.10a highlight the uncertainty derived from the twist angle error bar. While the two samples from the main text show an approximately similar $d_{\text{moiré}}$ (6nm and 4.5nm), the additional sample in the SN VI displays a much larger $d_{\text{moiré}}$ of 14nm.

Figure 10b shows the calculated edge of the mini-Brillouin zone ($\pi/d_{\text{moiré}}$) as a function of twist angle, the dots correspond to the specific samples studied in this work. Lastly, Fig. 10c shows the acoustic phonon dispersion relations for MoSe_2 and WSe_2 from Ref. [45]. The phonon dispersion is plotted in the region of small energies and wavevectors, with the energy and $\pi/d_{\text{moiré}}$ for the studied samples superimposed. While the sample stacked at 58.9° aligns within experimental errors with the TA and LA branches of the 1L-TMD materials, the samples with the larger twist angles clearly exhibit phonon modes that do not correspond to any vibrational branch. Although the previously mentioned renormalization processes may modify the phonon dispersion, this mismatch suggests that the vibrational modes involved in the polaron formation do not belong to the intrinsic 2D dispersion of these materials.

The discrepancy between the phonon dispersion relation and the energy and proposed wavevector of the observed phonon modes suggests a scalable reorganization of the monolayer crystal upon IX formation, leading to the emergence of localized vibrations that subsequently take part in the polaron formation. This process is schematically illustrated in Fig. 11, using a configurational coordinate diagram usually employed to present the Franck–Condon principle. Upon IX formation, the lattice undergoes a dynamic distortion which lowers the energy of the system. Thus, the IX is surrounded by the localized phonon cloud forming the polaronic state. As described in the main text and SN V, the phonon is a low energy mode whose occupation number n is greater than or equal to the thermal occupation number n_T . Therefore, during the HB emission process, the IX emits or absorbs phonons forming the emission phonon sidebands.

Within such a picture, the twist angle dependence observed in the phonon energy is then a measure of the crystal deformation within a moiré minimum in which an IX is confined. Consequentially, the samples shown in the main text, with similar moiré lattice parameter display a similar phonon energy. On the other hand, the sample presented in the SN VI host IXs in a larger moiré unit cell that requires a smaller crystal deformation, i.e., it requires a lower energy phonon to form polarons (see Fig.10a and c).

Figure 11: Configuration coordinate diagram illustrating the potential energy surfaces of the ground and excited states of a crystal lattice containing an IX, plotted as a function of a generalized lattice distortion coordinate. The two potentials are offset by a displacement parameter δ , representing the shift in equilibrium lattice configuration between electronic states. Quantized vibrational levels within each potential are separated by the phonon energy $\hbar\Omega$. The vertical optical transition between the lowest vibrational levels defines the ZPL at energy E_{ex} , while transitions involving additional phonon emission or absorption give rise to redshifted ($E_{\text{ex}} - N \hbar\Omega$) and blueshifted ($E_{\text{ex}} + N \hbar\Omega$) phonon sidebands, respectively.

The citations included in the Supplementary Material are:

[8] W.-T. Hsu, et al. ACS nano 8, 2951 (2014).

[17] K. L. Seyler et al. Nature 567, 66 (2019).

[18] T. Feldtmann, et al. Physica status solidi (b) 246, 332 (2009).

[19] T. Feldtmann, et al. Journal of luminescence 130, 107 (2010).

[24] W. Jin, et al. Nature communications 11, 4780 (2020).

[27] P. K. Nayak, et al. ACS nano 11, 4041 (2017).

[28] M. Kang, et al. Nature materials 17, 676 (2018).

[29] R. Brout, et al. Physical Review Letters 9, 54 (1962).

[30] A. Zaitsev. Physical Review B 61, 12909 (2000).

[31] M. Dyksik, et al. Advanced Science 11, 2305182 (2024).

[32] X. Zhang, et al. Chemical Society Reviews 44, 2757 (2015).

[33] P. Soubelet, et al. Physical Review B 93, 155407 (2016).

[34] B. Chakraborty, et al. Physical Review B—Condensed Matter and Materials Physics 85, 161403 (2012).

[35] Y. Wang, et al. Small 9, 2857 (2013).

[36] C. Qian, et al. Phys. Rev. Lett. 128, 237403 (2022).

[37] S. J. R. Tan, et al. Acs Nano 12, 5051 (2018).

[38] J. Quan, et al. Nature materials 20, 1100 (2021).

[39] M. Kögl, et al. Npj 2D Materials and Applications 7, 32 (2023).

[40] L. Li, et aql. Nature Communications 16, 4117 (2025).

[41] Z. Iakovlev, et al. Physical Review B 105, 205305 (2022).

[42] C. Li, et al. Nano Letters 23, 8186 (2023).

[43] S. Y. Lim, et al. Acs Nano 17, 13938 (2023).

[44] M. A. Semina, et al. Annalen der Physik 532, 2000339 (2020).

[45] der Physik 532, 2000339 (2020).

[45] M. Zulfiqar, et al. Scientific Reports 9, 4571 (2019).